PhyIN: trimming alignments by phylogenetic incompatibilities among neighbouring sites

Maddison Wayne P. wayne.maddison@ubc.ca
Beaty Biodiversity Museum and Departments of Zoology and Botany, University of British Columbia , Vancouver , British Columbia , Canada
Żyła Dagmara
Electronic publication date: 2024 Dec 5
Publication date: 2024
Volume: 12
Electronic Location ID: e18504
Received 2024 Jul 22; Accepted 2024 Oct 18
Copyright: ©2024 Maddison
Copyright year: 2024
Copyright holder: Maddison
License: This is an open access article distributed under the terms of the Creative Commons Attribution License, which permits unrestricted use, distribution, reproduction and adaptation in any medium and for any purpose provided that it is properly attributed. For attribution, the original author(s), title, publication source (PeerJ) and either DOI or URL of the article must be cited.
License URL: https://creativecommons.org/licenses/by/4.0/

Keywords: Phylogenomics, Alignment trimming, Phylogenetic inference, Software tool

Funding: NSERC Canada Discovery Grant This work was funded by an NSERC Canada Discovery Grant. The funders had no role in study design, data collection and analysis, decision to publish, or preparation of the manuscript.

==============================
In phylogenomics, regions of low alignment reliability and high noise are typically trimmed from multiple sequence alignments before they are used in phylogenetic inference. I introduce a new trimming tool, PhyIN, which deletes regions in which a large proportion of sites (characters) have conflicting phylogenetic signal. It does not require inference of a phylogenetic tree, as it finds neighbouring characters that cannot agree on any possible tree. In phylogenomic data of ultraconserved elements (UCE), PhyIN effectively finds the boundaries between chaotic (conflicted) and orderly regions of alignments with data for only a single locus. Its ability to work on individual loci allows it to preserve discord between gene trees and species trees.

Introduction

Phylogenomics typically relies on multiple sequence alignments to indicate what nucleotides (or amino acids) are homologous from sample to sample (taxon to taxon), thus allowing their evolutionary changes to be modeled while inferring a phylogenetic tree. However, those alignments can have regions of low confidence, their assessment of homology hindered by rapid evolution or errors in sequencing or assembly. For instance, alignments of loci recovered from genomic sequence-capture methods (e.g., Faircloth et al., 2012) are typically highly fragmented at their ends. Such highly fragmented and highly variable regions of an alignment, to the extent that they imply inaccurate homology or saturation, are expected to degrade or mislead phylogenetic inference (Ogden & Rosenberg, 2006; Wong, Suchard & Huelsenbeck, 2008; Duchêne et al., 2022). Even when saturated change does not confuse assessments of site homology, it may degrade phylogenetic inference. Thus, many phylogenomic pipelines apply trimming algorithms to delete doubtfully aligned or highly saturated parts of the alignment before phylogenetic analysis. Some, such as the often-used Gblocks (Castresana, 2000) and trimAl (Capella-Gutiérrez, Silla-Martínez & Gabaldón, 2009) delete whole sites (=columns, characters); others delete whole sequences (rows) (e.g., TreeShrink, Mai & Mirarab, 2018); others delete stretches within a single sequence that are outliers, perhaps because of mis-assembly (e.g., Spruceup, Borowiec, 2019; TAPER, Zhang et al., 2021).

The puzzle faced by trimming methods is how to identify noisy regions. Do they use an intuitive metric of disorder, or a model of the relevant biological rates and scientific error? Insofar as such trimming methods are gatekeepers for data subsequently analyzed by sophisticated model-based statistical methods, the trimming methods would ideally operate at that same level of sophistication. They would model evolutionary processes (e.g., mutation rates in the context of phylogeny) and errors from observation and inference methods (sequencing, assembly, alignment) to assess reliability of regions.

To date, no method attempts to trim using a full model of evolution and error. However, many trimming methods rely, at least loosely, on basic evolutionary principles. A critical principle is this: phylogenetic noise is not detectable in a single site (i.e., a single character, or column in the alignment). Phylogenetic noise is signaled by discord among characters. To illustrate why noise cannot be judged by characters examined individually, consider an individual-character state frequency criterion such that used by Gblocks (Castresana, 2000). If one character has nucleotide G in 9,999 sequences and C in one sequence, it would likely pass the test as conservative. Another character in which 5,000 have G and 5,000 have C would likely fail the test, which could lead to its deletion. One could argue that the 9,999:1 site is more conserved in function than the 5,000:5,000 site, but at stake in these analyses is evolutionary conservatism. The 5,000:5,000 character may be equally conserved in evolutionary terms. The 50:50 proportion of nucleotides may have arisen from a single evolutionary change marking a large clade of species with a distinctive nucleotide. Along the alignment there could be many such characters, if that clade and its sister clade are separated by long branches. One would not want those to be deleted by trimming—they contain the critical information to separate the two clades!—merely because they result in 50% of the species having one nucleotide and 50% another. To assess the 5,000:5,000 character’s reliability we would best compare its signal to that of other characters.

Methods that examine base frequencies at individual sites, but do not compare character patterns for discord, are therefore not designed to assess phylogenetic noise. Two examples of individual-site assessors are Gblocks (Castresana, 2000) and ClipKIT (Steenwyk et al., 2020). ClipKIT might appear to be phylogenetically designed, because of its focus on parsimony-informative characters, but this criterion is one of base frequencies, not of inter-character discord. The label “parsimony informative” has always been misleading, because its characters come with no guarantee of carrying information about phylogeny, merely that they are making a statement in parsimony terms (“parsimony-declarative characters” would be more apt). Their statement could be true or false, and indeed in genomic data most parsimony-informative characters are highly homoplasious and likely misleading (as seen in the data examined, below).

In a field that seeks both functional methods and conceptual advances, the value of a trimming method does not rest entirely on its demonstrable performance nor on its theoretical underpinnings. According to the theoretical stance I have outlined above, for instance, one should avoid methods like Gblocks and ClipKIT. However, in practice, they both appear to produce results that are reasonable (e.g., Steenwyk et al., 2020), despite their not being targeted to phylogenetic noise. This is much like the early rise of distance-based methods for phylogenetic reconstruction a half century ago. Although they had intuitive appeal, they were not based on a clear line of reasoning from a model of character change on phylogenetic trees to testable predictions (whether by parsimony or likelihood), and so in principle (many argued) they were best avoided. However, in practice, distance methods approximated phylogeny reasonably well. Nonetheless, focusing on what was correct in principle—explicit models of change on trees—was important for the subsequent development of the field, as it directed the field to move step by step towards the model-based approaches that are most broadly used today. Likewise, for trimming, it would be best to take steps, even if small, toward methods that are based more directly on phylogenetic logic, e.g., by looking for discord among characters.

Various trimming methods do compare characters, assessing whether the relationships suggested by one site or region are discordant with those suggested by the bulk of the data (e.g., Dress et al., 2008; Capella-Gutiérrez, Silla-Martínez & Gabaldón, 2009; Borowiec, 2019; Zhang et al., 2021). Ideally, the bulk-data relationships would take the form of a phylogeny on which to model rates and errors, but even an overall distance matrix can be a heuristic approximation of phylogenetic expectations (e.g., Capella-Gutiérrez, Silla-Martínez & Gabaldón, 2009; Borowiec, 2019).

The method I introduce here, PhyIN, is a small step forward. It is not based on a sophisticated evolutionary model, but on the basic phylogenetic principles of compatibility analysis (Le Quesne, 1969; Estabrook, Johnson Jr & McMorris, 1976; Meacham & Estabrook, 1985). Two characters are judged in conflict if they suggest different phylogenetic trees, that is, if there exists no tree for which both characters could have evolved without convergence. Pisani (2004) suggested deleting characters that are incompatible with many other characters in the data, but did not release a tool to implement the method. Dress et al. (2008) developed the Noisy method, which deletes characters that appear randomized by a compatibility criterion calculated on cyclic orderings of taxa. While the approaches of Pisani (2004) and Dress et al. (2008) assess a character’s compatibilities against all of the other characters, PhyIN looks for incompatibility among characters that are near neighbours in the alignment. This results in a remarkably simple trimming algorithm that nonetheless appears to distinguish chaotic versus ordered regions of alignments well in the data tested, better than Noisy, and better than trimAl done for individual loci.

PhyIN: Phylogenetic Incompatibility among Neighbours

The trimming method introduced here, PhyIN (“Phylogenetic Incompatibility among Neighbours”), seeks regions of the alignment in which a high proportion of characters are in phylogenetic conflict with one another. Phylogenetic compatibility of two characters can be assessed without reconstructing a phylogeny, because it depends on whether there exists a possible phylogeny on which the characters could agree (Meacham & Estabrook, 1985), that is, on which both of the characters can have no homoplasy (i.e., no convergence or reversal, i.e., no extra changes beyond the minimum conceivable). For two binary characters, the test for compatibility is simple. If some taxa have state 0 in the first character and state 1 in the second, other taxa 1 and 0, others 0 and 0, others 1 and 1—that is all four patterns are represented—then both characters cannot possibly agree on a phylogeny without homoplasy. They are incompatible. For two four-state characters (e.g., A, C, G, T), the test for incompatibility is only slightly more complex (Estabrook & McMorris, 1977; Semple & Steel, 2003). Draw two columns, one for the state in the first character (A, C, G, or T), and one for the state in the second character. Then, draw a line between the states in the two columns for each state combination observed among the taxa (e.g., a line between C in column 1 and G in column 2 reflects the existence of at least one taxon with state C in character 1 and G in character 2). Once all the combinations are drawn, there will be a graph with edges between the columns and vertices at each of the four states. If that graph has cycles, the two characters are incompatible (Semple & Steel, 2003).

The basic outline of the PhyIN method is:

1 Examine each character (column) in the alignment. If it is in conflict with a neighbour (i.e., phylogenetically incompatible as outlined above), mark it as conflicted.

2 Once conflicted characters are found, look for stretches or blocks of characters within which the proportion of conflicted characters is above a threshold.

3 If a block’s proportion of conflicted characters exceeds the threshold, the characters in that block are marked for deletion, from the block’s first conflicted to its last conflicted character.

4 Delete marked characters.

The settable parameters (with defaults indicated as of version 1.0) are:

-d (integer, default 2) Maximum distance of neighbours whose conflict is checked. In deciding if a character is conflicted, the method could look at just the immediately adjacent character, or characters up to -d columns away. Conflict is measured pairwise between characters, even if -d is greater than 1. As d increases, the trimming becomes more aggressive, but eventually would likely degrade PhyIN’s ability to find local boundaries between regions. In the UCE data I have examined, d = 2 appears to provide enough strength without eating into orderly regions.

-e (true/false, default true) Whether internal gaps are considered as a fifth (“extra”) state. Internal gaps are those separated from the end of the alignment by at least one observed nucleotide; terminal gaps are those in a contiguous stretch of gaps reaching to the sequence end. Terminal gaps in phylogenomic data are probably best interpreted as failure to sequence, not insertions/deletions. They should not be considered as a fifth state because they are not, in fact, gaps in the same sense as internal gaps, which do imply insertion/deletion events and therefore should be considered as seriously as nucleotide differences. For this reason, the default is e =true. Setting e =true will result in more aggressive trimming, but might best be restricted to trimming on separate loci. In concatenated alignments, the distinction between internal and terminal gaps is lost.

-b (integer, default 10) Size of blocks in which proportion of conflicted characters is counted. As the block size increases, there is more data by which to assess the noisiness of a region, but a short stretch of chaos might be missed (and not trimmed) if the blocks are too large. A value of 10 appears to work well for the UCE data tested. Note that the algorithm is not committed to trim the entire block; it trims from the first to last conflicted site only.

-p (proportion, 0 to 1, default 0.5) Threshold for proportion conflicted. Blocks whose proportion of conflicted characters is as great or greater than this will be deleted. Increasing this proportion makes the method more permissive, resulting in less trimming. Note that the number of conflicted characters can be larger than the number of conflicts, because a conflict is experienced by both of the two neighbours, or smaller than the number of conflicts, because there can be 6 pairwise conflicts among 4 conflicting characters.

Figure 1 shows an example with block size b = 10, distance d = 2, and threshold proportion p = 0.5. The block of 10 characters outlined by the rectangle is being assessed. Within that block, there are multiple conflicting characters. For instance, character 1 conflicts with character 2 because all four state combinations (TA, TT, AT, AA) are present (even though it is DNA data, these two characters are effectively binary, because each has only two variants). Character 1 conflicts also with 7 and 9, but the d = 2 parameter instructs us to look a maximum of two columns away for conflict. Thus, the counted conflicts are 1 vs. 2, 2 vs. 3, and 7 vs. 9. Characters 1, 2, 3, 7, and 9 are counted as conflicted, a total of 5 within the illustrated block of 10 contiguous characters. Because this proportion matches or exceeds the threshold of 0.5, the block is marked for deletion, but not the entire block—only from the first to last conflicted characters (1 through 9).

Figure 1 Trimming under the PhyIN criterion.

In the block outlined in the rectangle, 10 characters (sites) are being assessed (b = 10). Three conflicts are found among characters d = 2 or fewer sites apart. Because five characters of the 10 are involved in a conflict, this achieves the trimming threshold of p = 0.5. This results in a deletion of 9 contiguous characters (from first conflicted to last conflicted within the block of 10).

When internal gaps are treated as a fifth state (recommended for trimming on separate loci), PhyIN’s trimming is not much hindered by a modest proportion of internal gaps. Indeed, they may contribute to assessing conflict. However, regions consisting mostly of gaps (terminal or internal) will likely be poorly trimmed, because the few observed nucleotides give little opportunity for conflict.

Even though PhyIN’s criterion is based on phylogenetic incompatibility, passing or failing the criterion is not a complete assessment of phylogenetic information. As pointed out by a reviewer, characters that are in conflict in one part of the phylogeny could mutually support the tree in another part. In fact, that is a prime reason that parsimony methods (in which characters offer quantitative support for a tree) were long ago favoured over compatibility methods (by which a character as a whole can only vote yes or no). Nonetheless, a strong contrast in density of conflict among regions of an alignment suggests variation from region to region in phylogenetic informativeness. PhyIN therefore is expected to work best for data with such regional variation in informativeness.

PhyIN is not intended as a standalone solution to trimming. Most critically, because PhyIN does not address a high density of gaps per se (except as they contribute conflict as a fifth state), it should be combined with a filter for gappiness, and perhaps other filters as well. A gappiness filter need not assess phylogenetic reliability of characters, but can remove gappy regions simply because they have too little useful information to justify the computational cost and potential artifacts of including low-occupancy regions. In the examples shown below, I have used a simple filter that deletes any characters with more than 50% gaps (internal or terminal). In other words, it is an occupancy filter for sites just as many studies use a occupancy filter for loci (e.g., Azevedo, Hedin & Maddison, 2024; Castañeda-Osorio et al., 2024; Lin, Yang & Zhang, 2024).

Figure 2 shows how alignments of an ultraconserved elements (UCE) locus would be trimmed by PhyIN. The data come from Lin, Yang & Zhang (2024), for a group of salticid spiders of phylogenetic depth ca. 40 million years. PhyIN first selects sites for deletion using default settings (2B), then a site occupancy filter is applied (2C) leading the final trimming (2D).

Figure 2 PhyIN trimming of a UCE locus.

(A) Original untrimmed locus from spider UCE data of Lin, Yang & Zhang (2024). (B) Dark shows sites marked for trimming by PhyIN with default parameters (d = 2, e = true, b = 10, p = 0.5). (C) Dark shows sites marked for trimming by simple 50% site occupancy filter. (D) Combination of PhyIN and site occupancy filter trimming.

Comparisons among trimming methods

Trimming methods used in advance of phylogenetic analysis should be judged, ultimately, on the accuracy of the phylogenies they produce, but a complete assessment of accuracy requires known phylogenies in diverse groups (e.g., Tan et al., 2015) with various time depths, modes of evolution, and sequencing approaches. Simulations could be used (e.g., Capella-Gutiérrez, Silla-Martínez & Gabaldón, 2009), but they can fail to anticipate the complexities of evolution in inconveniently-behaving regions of genomes, which are a primary target of trimming. Considerable work remains to be done to assess trimming methods through known phylogenies or simulations. I will not attempt to fill that gap here. Nonetheless, some insight about trimming effectiveness can be gained using some simple statistics and intuitive visual inspection of results.

Visual inspection of trimmed alignments

Visual inspection relies on human pattern recognition to distinguish noisy from ordered sections of alignments. These distinctions need not correlate with phylogenetic informativeness, but a random-looking pattern of nucleotides gives little reason for confidence. Figure 3 offers examples for visual inspection, comparing the results of PhyIN with those of trimAl (Capella-Gutiérrez, Silla-Martínez & Gabaldón, 2009), Gblocks (Castresana, 2000) and Noisy (Dress et al., 2008), again with Lin, Yang & Zhang’s (2024) spider UCE data for several loci (loci chosen by being first in numerical order). Figures 3A–3F shows the results when loci are trimmed separately. PhyIN appears to cleanly find the boundaries between well aligned and chaotic regions (Figs. 3A and 3B). In contrast, trimAl (Figs. 3C and 3D), Gblocks (Fig. 3E), and Noisy (Fig. 3F) show more visibly chaotic regions (e.g., in Fig. 3C especially at start and end of loci 1 and 3). This “chaos” is not just an intuitive perception, because these regions were confirmed by PhyIN to have high levels of phylogenetic discord. When done on the concatenated alignments (Figs. 3G–3L), trimAl’s reduction of noise improves (Figs. 3I and 3J), presumably because the concatenation provided more data for trimAl’s similarity calculations. PhyIN’s results in Figs. 3A and 3G permit more gaps than trimAl’s, but that is an issue of the site occupancy filter rather than PhyIN. When that occupancy filter is set to 82%, which yields approximately the same total number of characters as the trimAl concatenation, PhyIN’s trimming appears no gappier than trimAl’s (Figs. 3B and 3H).

Figure 3 Comparison of trimming by PhyIN, trimAl, Gblocks, and Noisy.

First several loci in UCE data on salticid spiders (Lin, Yang & Zhang, 2024). In A–F, the loci were trimmed separately then concatenated; in G–L the loci were concatenated and then trimmed together. Gray lines show boundaries of loci. (A), (G) PhyIN at default settings (d = 2, e =true, b = 10, p = 0.5), then 50% site occupancy filter applied. (B), (H) PhyIN at default settings plus 82% site occupancy filter. The more stringent occupancy filter yields approximately the same total concatenated sequence length over the 3388 loci as trimAl in I and J, and thus a more direct comparison between the noise filters of PhyIN and trimAl. (C), (I) trimAl 1.4.1 with strictplus option. (D), (J) trimAl 1.4.1 with automated1 option. (E), (K) Gblocks 0.91b at default settings except -b5 =h to permit some gaps. (F), (L) Noisy 1.5.12, at cutoff value 0.99, which appeared best in cleaning up noise but retaining data.

Noisy (Dress et al., 2008) was chosen for comparison because it is based on the same concept as PhyIN, character compatibility. Its assessment of a character’s incompatibilities across the entire alignment appears to result in less effective trimming than PhyIN’s local comparisons, as Noisy not only consumed much of the apparently well-aligned areas, but left much noise (Figs. 3F and 3L).

To explore the different trimmings of PhyIN and trimAl, characters that were retained by PhyIN but trimmed by trimAl, and those retained by trimAl but trimmed by PhyIN, were collected by Mesquite (Maddison & Maddison, 2024) for the Lin, Yang & Zhang (2024) salticid data. The results are shown in Fig. 4. The columns in the PhyIN-retained data appear visibly cleaner, and offer no obvious reason to be discarded (except for rogue subsequences, which appear more commonly in the trimAl-retained parts, and which can be removed as mentioned below). While the cleanliness of sites retained by PhyIN might imply lack of useful variation, the speckled appearance of sites retained by trimAl suggests noise. This is not an issue of rows being in a haphazard sequence, because they were reordered to match the sequence of terminals in the phylogeny derived from the trimAl data, in order maximize the appearance of useful variation in the trimAl alignment. (This comparison was designed to benefit trimAl’s case, with PhyIN run locus-by-locus and trimAl using concatenated data).

Figure 4 Comparison of sites kept or discarded by PhyIN versus trimAl.

First several loci in UCE data on salticid spiders (Lin, Yang & Zhang, 2024). (A) Sites kept by PhyIN but discarded by trimAl -strictplus. PhyIN run at default settings and with site occupancy filter at 82%. The 82% filter yields approximately the same total concatenated sequence length over the 3,388 loci as trimAl, and thus a more direct comparison of noise filters of PhyIN versus trimAl. (B) Sites kept by trimAl-strictplus but discarded by PhyIN.

Diagnostic statistics

To explore alternative trimming methods, five published datasets using UCEs in animals were examined quantitatively (Fig. 5). Three datasets represent salticid spiders at various phylogenetic depths. The “salticid-shallow” dataset concerns harmochirines (from Azevedo, Hedin & Maddison, 2024; 48 taxa, 279 loci, ingroup ∼15 million years deep) (one taxon, Havaika sp. d300, was removed because it had data for only 32 loci). The “salticid-mid” dataset is the one highlighted in Figs. 2–4, concerning the Salticoida (from Lin, Yang & Zhang, 2024; 46 taxa, 3388 loci, ingroup ∼38 million years deep, Bodner & Maddison, 2012). The “salticids-deep” dataset concerns the whole family Salticidae (from Zhang et al., 2024; 70 taxa, 3,267 loci, ingroup ∼50 million years deep, Bodner & Maddison, 2012). The three salticid datasets are largely independent sets of taxa, with the salticids-shallow representing only a small portion of the diversity in the salticids-mid, and likewise the mid in the deep. The wasp dataset concerns the doryctine braconids (from Castañeda-Osorio et al., 2024; 713 loci, 161 taxa, early Cenozoic divergences, Zaldivar-Riverón et al., 2008). The bivalve dataset concerns all bivalve molluscs (from González-Delgado, Rodrígues-Fores & Giribet, 2024; 757 loci; 59 taxa; ingroup diverged early Paleozoic). The authors of these datasets supplied alignments of loci intercepted from their pipelines just before they were trimmed. From each dataset, loci were removed if they were present in fewer than 50% of the taxa. For each of these datasets, the original untrimmed alignments and varied trimmings are available on Figshare at doi: 10.6084/m9.figshare.27122211.

Figure 5 Comparison of trimmers with five UCE datasets.

Datasets shown in approximate sequence from more recent to older diversifications. Trimming done either on separate loci before concatenation (dark grey bars) or after concatenation (light grey bars). (A) Proportion of characters (sites) trimmed. (B) Among the parsimony-informative characters, proportion that are concordant with the tree (the others show homoplasy). Datasets are salticids-shallow (Harmochirina, divergences late Cenozoic), salticids-mid (Salticoida, mid Cenozoic), salticids-deep (Salticidae, early-mid Cenozoic), braconid wasps (Doryctinae, early Cenozoic), and bivalve molluscs (early Paleozoic). See text for references.

I trimmed each dataset by PhyIN, trimAl, and Gblocks. For each method, trimming was done both before concatenation (locus by locus) and after concatenation (dark grey versus light grey in Fig. 5). PhyIN trimming was accompanied by a simple filter to remove sites with less than 50% occupancy (i.e., more than 50% gaps), as discussed above. Trimming by trimAl was done with both the automated1 and strictplus options. Gblocks was performed with default values except that 50% gaps were permitted (-b5 =h). In addition, as a baseline, statistics were examined under a weak trim that made no attempt to remove noisy sites, using only the site occupancy criterion (“50%s.o.”) to remove sites with more than 50% gaps.

Figure 5A shows the first of the two statistics examined, the proportion of sites trimmed. In each dataset, PhyIN trims about as many sites whether done locus by locus or on the concatenated alignment. In contrast, trimAl trims distinctly fewer sites when done by locus. Although trimming intensity is not a direct reflection of quality, examination of trimmings suggests trimAl’s weaker trimming on separate loci likely reflects its failure to remove some noisy regions when given only a single locus (as seen in Figs. 3F versus 3G). Gblocks likewise shows more aggressive trimming on the concatenated alignment, but that appears to reflect its treatment of gaps rather than of noise. Locus by locus, Gblocks is aware only of the taxa represented by sequences for a gene, and so does not count a taxon’s absence in that gene as gaps, but in the concatenated alignment those gaps are counted. In general, PhyIN (with the 50% site occupancy filter) is approximately as aggressive in trimming as trimAl or default Gblocks.

Figure 5B shows a crude measure of phylogenetic signal in the trimmed data: the proportion of parsimony-informative sites that are concordant with the phylogeny as inferred by RAxML (unpartitioned) on those same data. Those concordant characters show unambiguous synapomorphies for a clade and no convergences, while the other parsimony-informative characters (more than 75% of them) show homoplasy. This metric could be problematical, not only because it is not based on stochastic models, but also because a character could offer noise in one part of a phylogeny but useful signal in another. Nonetheless, the patterns seen in Fig. 5B appear to correlate well with the visual noise seen in Fig. 3. PhyIN gave higher proportions of concordant sites in all datasets except wasps, where trimAl on the concatenated data showed relatively more concordant sites. When done on separate loci, however, trimAl consistently resulted in higher proportions of homoplasious sites, especially in the wasp and two deeper salticid datasets.

The phylogenies obtained by RAxML (Stamatakis, 2014, unpartitioned) from concatenated alignments trimmed by these various methods were little different for the spider datasets, but more disparate for the wasps and bivalves. The nine methods (Gblocks, trimAl-strictplus, trimAl-automated1, and PhyIN each done before and after concatenation, plus 50%s.o.) yielded trees that differed in up to seven clades for the salticids-shallow dataset, up to three for the salticids-mid dataset, up to two for salticids-deep, up to 35 for wasps, and up to 24 for bivalves. The wasp dataset has many more taxa, and hence many more clades that could differ. In all datasets the PhyIN trimmings done before and after concatenation yielded more similar trees (one, zero, zero, 10, zero clades different for the same five datasets, respectively) than the before versus after trimmings of trimAl strictplus (one, two, two, 32, two clades different), of trimAl automated1 (five, two, two, 32, one), or of Gblocks (two, one, zero, 22, 23).

Trimming loci separately vs. concatenated

PhyIN is efficient in using information in a single locus: its trimmings differ little whether done locus by locus or on the concatenated alignment. The black and grey bars of Fig. 5 are nearly the same for PhyIN whether by locus or concatenated, the trees recovered are little different, and the apparent visual noise is similar and low for both (Figs. 3A vs. 3G). In contrast, the other trimming methods generally show weaker trimming (Fig. 5A) and higher relative homoplasy (lower concordant sites, Fig. 5B) for the locus-by-locus trimming compared to the concatenated, stronger differences in the trees, and distinctly worse visual noise for the by-locus trimming (Figs. 3C–3E vs. 3I–3K). PhyIN’s consistency is not a matter of consistent failure. Its proportion of concordant sites (Fig. 5B) and visual noise (Fig. 3) match or are better than other methods’ in most datasets.

PhyIN’s consistency of results whether used on loci separately or concatenated is to be expected, because its choice to delete a site depends only on other sites nearby. What is surprising (it was to me) is that it trims so well with such limited information, and with such a compact algorithm (<100 lines of Python code in the core calculation).

PhyIN’s success is not likely the result of some fundamental evolutionary principle, but rather of the structure of the genomic regions studied: the data contain contrasting regions of high noise versus high signal which are therefore distinguishable by local comparisons among neighboring characters. Data from UCEs is expected to be structured like this. Whether PhyIN does as well with data from other sequencing methods (e.g., low coverage genomes) remains to be studied.

The ability of PhyIN to trim well with individual loci is advantageous for phylogenomic studies that investigate or account for conflict between gene trees. Different regions of the genome may have discordant phylogenetic histories, and thus discordant patterns of similarity, because of incomplete lineage sorting, introgression, or lateral gene transfer (Maddison, 1997). For studies of species trees and gene trees, that discord is part of the signal, not noise. A trimming tool applied over a concatenated alignment could label as noisy those loci that disagree with the overall similarities of the majority of loci. The signal from those discordant loci could therefore be trimmed, silencing their voices and obscuring discordant processes. Thus, to the extent possible, it is better to trim loci separately for such studies.

Implementation and use in a trimming pipeline

PhyIN is designed as a module in Mesquite (see Maddison & Maddison, 2023) that flags sites as trimmable, which can then be visualized, trimmed, or excluded (without trimming) for analyses. It is available currently in the development branch of Mesquite (Maddison & Maddison, 2024; see http://www.mesquiteproject.org/Source%20Code.html for installation instructions) and will be available in the forthcoming stable release version 4.0. In Mesquite’s matrix editor window, the relevant menu items are (1) Alter>Trim Sequences>Trim Sites by PhyIN; (2) Display>Color Matrix Cells>Highlight by Trimming or Flagging Methods; (3) Select>Select Characters>Select by PhyIN; and (4) Matrix>Character Inclusion/Exclusion>Add (or Replace) Exclusions by Trimming Method. In the list of character matrices window, go to Utilities>Alter Matrices. PhyIN trimming is also available via File>Open Other>Process Data Files... under Add Step, Alter Matrices. This version of Mesquite also integrates access to trimAl, Gblocks, and a subset of Spruceup (Borowiec, 2019) to allow users to quickly compare or combine trimming options. The Java code within Mesquite that does the PhyIN calculations is available on GitHub (https://github.com/wmaddisn/PhyIN, DOI: 10.5281/zenodo.14195271). For use in scripted pipelines, PhyIN is available also in that same GitHub repository as a standalone Python script, which includes a simple filter for gappiness of sites (the site occupancy criterion). Current versions are restricted to nucleotide (not amino acid) data.

PhyIN is not intended to be a complete and sufficient trimming system, but rather an additional filter in the phylogenomicist’s toolkit. A suggested sequence of steps is this:

1 Run PhyIN locus-by-locus. The rationale of doing PhyIN before other trimming or concatenation is that it can then accurately distinguish between internal and external gaps.

2 Run a filter to remove gappy sites or blocks (e.g., Fig. 2B). For this study I have used the simple 50% site occupancy filter in Mesquite after all the loci are within a single file, whether matrices are separate or concatenated, because then Mesquite knows the total number of taxa in the study. (The filter would be less stringent if done on each locus as an isolated file, as then the trimmer would be unaware of the other taxa lacking data for the locus).

3 Concatenate the loci into a single alignment.

4 Run Spruceup (Borowiec, 2019) to remove outlier subsequences. In all of the trimmings there appear to be some “rogue” subsequences, a short stretch of nucleotides in one taxon whose distinctiveness is out of proportion with the overall distinctiveness of that taxon based on other parts of the data, perhaps as a result of misassembly or genomic rearrangements. For example, in Figs. 3A–3E there are rogue subsequences toward the end of the first locus in the 8th taxon and in the taxon 5th from last, and another in the middle of the first locus in the taxon 9th from last. With the data tested here, Spruceup (Borowiec, 2019) appears to have high specificity in cleaning those rogue subsequences, but only when used on concatenated data rather than locus by locus. While there may be concerns that its application on the concatenated data could obscure discordant gene histories as discussed above, it appears that Spruceup’s trimming is isolated to only those subsequences that appear fully random compared to the others. Detailed consideration of Spruceup’s behaviour, however, is outside the scope of this paper.

5 Deconcatenate the loci to separate alignments if individual gene trees are desired. This can be done in Mesquite (Maddison & Maddison, 2024) if the previous steps were run within Mesquite.

Supplemental Information

Supplemental Information 1 Python code for PhyIN alignment trimming

This code performs PhyIN trimming on a fasta file. For details as to how to use at the command line, see comments near top of code.

I thank Mike Steel for guidance about multistate compatibility tests. J. Zhang, G. Azevedo, R. Castañeda-Osorio, A. Zaldívar-Riverón, S. González-Delgado, and G. Giribet generously supplied the original alignments of their UCE data. Marek Borowiec and two anonymous reviewers provided helpful comments on a previous version of this manuscript. Marek Borowiec, Junxia Zhang, Rodrigo Monjaraz Ruedas, Brant Faircloth, David Maddison, and Marshal Hedin all offered helpful discussion and advice.

Additional Information and Declarations

Competing Interests

Author Contributions

Data Availability

The author declares that he has no competing interests.

Wayne P Maddison conceived and designed the experiments, performed the experiments, analyzed the data, prepared figures and/or tables, authored or reviewed drafts of the article, and approved the final draft.

The following information was supplied regarding data availability:

The code for PhyIN is available at GitHub and Zenodo (both in Java and Python)

- https://github.com/wmaddisn/PhyIN

- Wayne Maddison. (2024). wmaddisn/PhyIN: PhyIN v.0.992 (v0.992). Zenodo. https://doi.org/10.5281/zenodo.13851761.

The Python code is standalone, but the Java is a snippet that depends on the development version of Mesquite, which is available at Github: https://github.com/MesquiteProject/MesquiteCore/tree/development.

The example data used in the figures is available, both untrimmed and trimmed, in Figshare: Maddison, Wayne (2024). Example datasets used by W. Maddison in ”PhyIN: Trimming alignments by phylogenetic incompatibilities among neighbouring sites”, submitted to PeerJ 2024. figshare. Dataset. https://doi.org/10.6084/m9.figshare.27122211.v1.

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
