# Peer review of "PhyIN: trimming alignments by phylogenetic incompatibilities among neighbouring sites"

_PeerJ, doi:10.7717/peerj.18504_

## Round 0.1 · original submission · Major Revisions

Please, answer all reviewers' comments and take their suggestions into account. Whenever you disagree with a suggestion, clearly explain your reasoning.

Reviewer 1 ·

Basic reporting

The article uses clear language and and technically correct text throughout. The article conforms to professional standards for published research articles. The mostly article uses appropriate literature references and provides sufficient background and context for the work. One exception is the absence of reference to similar recent work, describing how removing saturated sites showing phylogenetic discordance can improve phylogenetic inference (Duchene et al. 2022; https://academic.oup.com/sysbio/article/71/3/676/6368692). This arguments are very similar to the work presented here, and so this should be cited.

The work is self-contained with relevant results to hypotheses.

Experimental design

Major Comment 1: the key concept of this work is that alignments can be improved by removing sites that are incompatible in terms of phylogenetic signal (i.e., they cannot be mapped onto the same tree without a conflict that requires homoplasies or other non-parsimonious processes to explain). Homoplasies, misalignment, and error are all treated the same by this approach. Phylogenies generated using alignments trimmed in this way will therefore be more accurate, having less input from discordant sites. This approach is also extremely efficient, as it does not require constructing a phylogeny from a whole alignment, as the signal within individual sites is compared in a non-phylogenetic way. However, an important assumption being made appears to be that that sites containing homoplasies contribute to phylogenetic conflict more than consensus, at least, to the point where removing them will improve phylogenetic inference from the resulting trimmed alignment. This seems to be an unfounded assumption on two points:

Point 1: individual sites may be incompatible with the "true" tree or neighboring sites for some bipartitions, while still both supporting other bipartitions. That is, different sites can provide meaningful support to different branches within the same tree, while being "noisy "with respect to other branches. Hypothetically, an alignment could exist where every site contains an incompatible phylogenetic signal, while overall every site contributes to a strong consensus support for a single tree and a single set of bipartitions.

Point 2: even sites with sequence saturation that have eroded phylogenetic signal through homoplasies meaningfully contribute to maximum-likelihood tree inference, as they inform branch lengths within the tree, even if they have a similar likelihood under different topologies. Therefore, removing sites that are in well-aligned regions of sequence because they contain conflicting phylogenetic signals may result in a mis-estimation of branch lengths, and thus the overall maximum-likelihood function.

A phylogeny-based metric to ascertain the impact of this trimming approach on tree topology and likelihood should be employed to test both points, as well as to provide quantitative metrics for Major Comment 2 (below).


Major Comment 2:

The work relies heavily on comparison of PhyIN to other trimming sites. However, the comparisons made are intuitive and qualitative (e.g., line 203, "remarkably similar"); the authors do not provide a quantitative or objective metric comparing the output of each trimming approach. This could include, number of sites trimmed, quantifying overlaps between approaches, and phylogenetic analyses of resulting trimmed alignments to assess likelihood, tree similarity, or other tree-based metrics of the efficacy of trimming. Without these tests, it is difficult ascertain if PhyIN offers an improvement over existing methods.

Validity of the findings

no comment

Additional comments

The authors make a timely and valuable observation that existing trimming algorithms often conflict with one another, and fail to make use of phylogenetic hypotheses. The shortcomings of these techniques are well-reasoned and clearly shown in the text and figures.



Minor comments:

Introduction: The authors state that:

"...alignments can have regions of low confidence, their assessment of homology hindered by rapid
evolution or errors in sequencing or assembly."

However, this is true for rapid evolution only if it introduces indels. Rapid evolution at a site can saturate the site, resulting in a loss of phylogenetic information, but the homology of the characters across sequences at this site is not brought into question. The authors should make this more clear.

Reviewer 2 ·

Basic reporting

The author meets the guidance for this section.

Experimental design

The author meets the guidance for this section.

Validity of the findings

The author meets the guidance for this section.

Additional comments

See attached file.

Annotated reviews are not available for download in order to protect the identity of reviewers who chose to remain anonymous.

---

## Round 0.2 · accepted · Accept

All the reviewers' comments were sufficiently addressed. I am happy to accept the current version of the manuscript for publication.

Reviewer 1 ·

Basic reporting

The author has sufficiently responded to my previous comments and has updated the manuscript and submission materials accordingly.

Experimental design

The author has sufficiently responded to my previous comments and has updated the manuscript and submission materials accordingly.

Validity of the findings

The author has sufficiently responded to my previous comments and has updated the manuscript and submission materials accordingly.

Additional comments

The author has sufficiently responded to my previous comments and has updated the manuscript and submission materials accordingly.

Reviewer 2 ·

Basic reporting

The author has addressed all my comments and concerns.

Experimental design

The author has addressed all my comments and concerns.

Validity of the findings

The author has addressed all my comments and concerns.

Additional comments

The author has addressed all my comments and concerns.